# Expectation Programming: Adapting Probabilistic Programming Systems to Estimate Expectations Efficiently

**Tim Reichelt**[1]      **Adam Goliński**[1]      **Luke Ong**[1]      **Tom Rainforth**[1]

[1]University of Oxford

## Abstract

We show that the standard computational pipeline of probabilistic programming systems (PPSs) can be inefficient for estimating expectations and introduce the concept of *expectation programming* to address this. In expectation programming, the aim of the backend inference engine is to directly estimate expected return values of programs, as opposed to approximating their conditional distributions. This distinction, while subtle, allows us to achieve substantial performance improvements over the standard PPS computational pipeline by tailoring computation to the expectation we care about. We realize a particular instance of our expectation programming concept, Expectation Programming in Turing (EPT), by extending the PPS *Turing* to allow so-called *target-aware* inference to be run automatically. We then verify the statistical soundness of EPT theoretically, and show that it provides substantial empirical gains in practice.

## 1 INTRODUCTION

Estimating *expectations* is at the center of many scientific workflows. For example, the decision theoretic foundations of most statistical paradigms, e.g. Bayesian decision theory, are rooted in calculating the expectation of a loss function [Robert and Casella, 2004].

Carrying out this estimation often requires *approximate inference* to be performed: we may not be able to directly draw samples of the random variable we wish to calculate the expectation of, or a simple Monte Carlo estimate might produce problematically high variance.

Probabilistic programming systems (PPSs) provide a powerful basis for encoding such inference problems and then assisting with, or even fully automating, the approximation

of their solution [Gordon et al., 2014, van de Meent et al., 2018]. In a PPS, programs are typically specified (often indirectly) through an unnormalized density $\gamma(x)$. Assuming analytic solutions are not available, the role of the system's inference engine is now to construct an approximation, $\hat{\pi}(x)$, for the distribution specified by the normalized density $\pi(x) = \gamma(x)/Z$, where $Z$ is an unknown normalizing constant and $\pi(x)$ typically represents a conditional distribution, such as the posterior in a Bayesian modeling setting. This approximation can then be used in turn for downstream tasks, such as approximating one or more expectations.

Though ostensibly very general, our key insight is that this standard PPS computational pipeline—which is implicitly followed by all contemporary PPSs that conduct inference approximately (e.g. Bingham et al. [2019], Carpenter et al. [2017], Cusumano-Towner et al. [2019], Ge et al. [2018], Salvatier et al. [2016], Tran et al. [2016], Wood et al. [2014], Mansinghka et al. [2014], Goodman and Stuhlmüller [2014], Murray and Schön [2018], Minka et al. [2018])—can be highly suboptimal when our ultimate aim is to estimate a particular expectation, $\mathbb{E}_{\pi(x)}[f(x)]$. This is because such a pipeline fails to perform estimation in a *target-aware* fashion: it does not allow information about $f$ to be exploited by the inference engine, thereby forgoing the substantial empirical gains that using information about $f$ can yield [Torrie and Valleau, 1977, Hesterberg, 1988, Wolpert, 1991, Oh and Berger, 1992, Evans et al., 1995, Meng and Wong, 1996, Chen et al., 1997, Gelman and Meng, 1998, Lacoste-Julien et al., 2011, Owen, 2013, Golinski et al., 2019, Rainforth et al., 2020]. Note here that it is not generally possible to incorporate the required information about $f$ by adjusting the model definition; fundamental changes to the computational pipeline itself are required.

To address this, we introduce, and formalize, the concept of *expectation programming*. Here an expectation program is analogous to a probabilistic program, but its target quantity of interest is the expected value of the program's return values, rather than their conditional distribution. This subtle distinction leads to changes in the requirements for the pro-

*Accepted for the 38th Conference on Uncertainty in Artificial Intelligence* (UAI 2022).

gram to be valid, and, critically, the estimation that must be performed by the backend inference engine. This, in turn, allows us to construct computational pipelines which are target-aware, utilizing information in the program itself to estimate expectations substantially more efficiently than can be achieved by existing PPSs.

We realize our expectation programming concept through a specific system we call **EPT** (Expectation Programming in Turing), built upon the Turing PPS [Ge et al., 2018]. EPT takes as input a Turing-style program and uses a combination of program transformations and existing inference strategies to construct target-aware estimators via the TABI approach of [Rainforth et al., 2020].

We formally demonstrate the statistical soundness of EPT, proving that it produces consistent estimates under nominal assumptions. We further show empirically that it can be used to express and run effective inference for a number of problems, finding that it produces estimates that are significantly more accurate than conventional usage of Turing. As part of this, we also implement a new annealed importance sampling (AnIS) [Neal, 2001] inference engine for Turing, finding that this allows for effective marginal likelihood estimation in a much wider array of problems than Turing's previously supported inference strategies.

To summarize, our key contributions are: a) identifying the shortfall of existing PPSs when estimating expectations and introducing the concept of expectation programming to address this; b) developing EPT as a particular realization of the expectation programming concept; c) formalizing the notion of an expectation program and demonstrating the statistical correctness of EPT; d) introducing a new AnIS inference engine to Turing; and e) showing that EPT can provide substantial empirical benefits over conventional use of Turing on real problems.

## 2 BACKGROUND

### 2.1 TURING PROGRAMS AS DENSITIES

To provide a basis for introducing expectation programming, we consider the PPS Turing (Ge et al. [2018], https://turing.ml/dev/docs/using-turing/), but note that the concepts introduced apply to PPSs in general. We provide a brief introduction to Turing here, along with our own new formalism for the densities Turing program define by extending the approach of Rainforth [2017, §4.3]. This is necessitated by some technical intricacies of the expectation programming approach. To assist with this, we will use the following simple Turing program as a running example:

```julia
@model function model(y)
    x ~ Normal(0, 1)
    @addlogprob!(0.1)
    y ~ Normal(x, 1)
end
```

A Turing program is defined similarly to a normal Julia function [Bezanson et al., 2017]: the `@model` macro indicates the definition of a Turing model, with tilde statements inside the body, e.g. `x ~ Normal(0, 1)`, to denote probabilistic model components. Observed data can be passed in as a formal argument to the function. If the variable name on the left-hand side of the tilde statement is not part of the arguments of the functions then it is interpreted as a random variable.

Let $x_{1:n}$ denote the set of direct outputs from sampling statements and $y_{1:m}$ the observed data. We can view Turing programs as defining an unnormalized density $\gamma(x_{1:n})$ (with an implicit appropriate reference measure). To compute the density for a given $x_{1:n}$ the program executes like a normal Julia program, while keeping track of the density of the current execution. Specifically, when Turing reaches a tilde statement corresponding to a random variable, it samples a value for $x_i$, evaluates the density of this draw, and factors this into the overall execution density. We denote the density of the draw as $g_i(x_i|\eta_i)$, where $g_i$ denotes the form of the sampling statement and $\eta_i$ its parameters. For the tilde statements corresponding to the observed data, it evaluates the density function $h_j(y_j|\phi_j)$—where $h_j$ and $\phi_j$ are analogous to $g_i$ and $\eta_i$ respectively—and factors the overall density accordingly.

Sometimes a user might want to add additional factors to the density without using a tilde statement. For this, Turing provides the `@addlogprob!(log_p)` primitive which multiplies the density of the current execution by an arbitrary value `exp(log_p)`. We use $\psi_1, \ldots, \psi_K$ to denote all the terms that are added to the density using `@addlogprob!`.

Putting these together, the unnormalized density defined by any valid program trace can be written as

$$\gamma(x_{1:n}) = \prod_{i=1}^{n} g_i(x_i|\eta_i) \prod_{j=1}^{m} h_j(y_j|\phi_j) \prod_{k=1}^{K} \exp(\psi_k). \quad (1)$$

Our example program thus defines the density $\gamma(x) = \exp(0.1)\mathcal{N}(x; 0, 1)\mathcal{N}(y; x, 1)$, with a fixed input $y$. Note here that everything (i.e. $n, x_{1:n}, \eta_{1:n}, g_{1:n}, m, y_{1:m}, \phi_{1:m}, h_{1:m}, K, \psi_{1:K}$) can be a random variable because of potential stochasticity in the program path. However, using the program itself, everything is deterministically calculable from $x_{1:n}$, which can thus be thought of as the 'raw' random draws that dictate all the randomness of the program; everything else is a pushforward of these.

### 2.2 TARGET-AWARE INFERENCE

Consider the problem of estimating an expectation of the form $\mathbb{E}_{\pi(x)}[f(x)]$ where $f(x)$ is known, but $\pi(x)$ cannot be directly evaluated or sampled from. Namely, $\pi(x) = \gamma(x)/Z$ where $\gamma(x)$ is a known unnormalized density, but $Z$

is an unknown normalization constant (sometimes referred to as the marginal likelihood or partition function).

The inference engines in PPSs like Turing are setup to approximate $\pi(x)$ of this form. As such, the standard pipeline to approximate an expectation using a PPS is to first approximate $\pi(x)$ (e.g. with samples) and then use this to approximate the expectation in turn.

Unfortunately, this ignores information about $f$ and is therefore suboptimal if $f$ is known [Golinski et al., 2019]. While one might initially expect that information about $f$ can be easily incorporated through simple model adjustments, this is unfortunately not the case in practice: any adjustments we make will mean we need to estimate an additional corrective factor on top of performing inference for the new model. Indeed, naive approaches to incorporating information about $f$, like adding $|f(x)|$ as a density factor to the model, have been found to typically worsen, rather than improve, the final estimates [Rainforth et al., 2020].

Rainforth et al. [2020] recently showed that this issue stems from fundamental limitations of the efficacy of using a *single* Monte Carlo estimator for such expectations. Namely, through their Target-Aware Bayesian Inference (TABI) framework, they show that by breaking down the expectation into three parts:

$$\mathbb{E}_{\pi(x)}[f(x)] = (Z_1^+ - Z_1^-)/Z_2, \qquad (2)$$

where $Z_1^+ = \int \gamma(x)f^+(x)dx$, $Z_1^- = \int \gamma(x)f^-(x)dx$, $Z_2 = \int \gamma(x)dx$, $f^+(x) = \max(f(x), 0)$, and $f^-(x) = -\min(f(x), 0)$, and then estimating each term separately, one can often achieve a substantially improved overall estimator, $\mathbb{E}_{\pi(x)}[f(x)] \approx (\hat{Z}_1^+ - \hat{Z}_1^-)/\hat{Z}_2$.

The intuition here is that each individual term can often be estimated more accurately in isolation than the original expectation. To see this, first note that the three subcomponents can be seen as the respective normalization constants of the three densities

$$\begin{aligned}
\gamma_1^+(x) &\propto \gamma(x)f^+(x), \\
\gamma_1^-(x) &\propto \gamma(x)f^-(x), \qquad (3) \\
\gamma_2(x) &= \gamma(x).
\end{aligned}$$

The TABI framework now allows one to define a separate estimator *tailored* to each of these problems. In general, it allows one to repurpose any algorithm which provides estimates of the normalization constant into a target-aware inference algorithm by separately applying it to each of $\gamma_1^+(x)$, $\gamma_1^-(x)$, and $\gamma_2(x)$. TABI can theoretically achieve an arbitrarily low error for any fixed sample budget ($\geq 3$), unlike standard approaches such as self-normalized importance sampling or MCMC whose expected error is lower bounded, even when using an optimal proposal/sampler. The achievable gains increase, both theoretically and empirically, with the degree of mismatch between $\pi(x)$ and $\pi(x)f(x)$.

# 3 EXPECTATION PROGRAMMING

At a high level, *expectation programming* adapts probabilistic programming systems to automate the *estimation* of expectations in a *target-aware* manner. As we now explain, an *expectation program* is analogous to a probabilistic program, but where the quantity of interest is the expectation of its return values under the program's conditional distribution, rather than the conditional distribution itself.

## 3.1 FORMALIZATION

To formalize the concept of an expectation program, we first statistically formalize probabilistic programs as follows.

**Definition 1.** *A probabilistic program $\mathcal{P}$ in a probabilistic programming language defines an unnormalized density $\gamma(x_{1:n})$ over the raw random draws $x_{1:n} \in \mathcal{X}$ of the program, which collectively we refer to as the program trace, along with an implicitly defined reference measure $\mu$.*

We let $\pi(x_{1:n}) = \gamma(x_{1:n})/Z$ denote the normalized density with $Z = \int_{\mathcal{X}} \gamma(x_{1:n})d\mu(x_{1:n})$. Here $\pi(x_{1:n})$ and $\mu$ combined implicitly define the conditional probability distribution specified by $\mathcal{P}$, which we denote $\mathbb{P}(A) = \int_A \pi(x_{1:n})d\mu(x_{1:n})$.

To ensure that the induced probability measure of a program is well-defined, we require that $\gamma(x_{1:n})$ corresponds to a valid unnormalized density. This guarantees that there is a valid probability distribution the inference algorithm of the particular PPS can converge to. We use this to formalize the concept of a *valid* probabilistic program as follows.

**Definition 2.** *A probabilistic program, $\mathcal{P}$, is valid (and defines a valid unnormalized probabilistic program density $\gamma(x_{1:n})$) if and only if both of the following hold: $\gamma(x_{1:n}) \geq 0, \forall x_{1:n} \in \mathcal{X}$; and $0 < \int_{\mathcal{X}} \gamma(x_{1:n})d\mu(x_{1:n}) < \infty$.*

For Turing we have described how programs specify $\gamma(x_{1:n})$ in Section 2.1, but Definitions 1 and 2 apply more generally and only require that we can derive an unnormalized density function for a given program; a requirement that is satisfied by most existing popular PPSs.

We can now formalize the concept of an expectation program by associating return values to our program:

**Definition 3.** *An expectation program, $\mathcal{E}$, is a probabilistic program (as per Definition 1) with an associated set of return values $F \in \mathcal{F} \subseteq \mathbb{R}^d$ that are a deterministic mapping of the trace $x_{1:n}$.*

From this definition we see that expectation programs are largely equivalent to probabilistic programs, indeed programs in any PPS that allows return values will also be expectation programs provided their outputs are numeric and fixed dimensional. However, as their underlying quantity of interest is the expectation of their return values, $\mathbb{E}[F]$,

they require a slightly different set of assumptions to ensure validity as follows.

**Definition 4.** *An expectation program $\mathcal{E}$ is valid if and only if it is a valid probabilistic program and $F$ is integrable.*

Here the additional requirement of the expectation program's outputs being integrable essentially equates to requiring that the expectation $\mathbb{E}[F]$ exists and $\mathbb{E}[|F_i|] < \infty$ for each dimension $F_i$ of $F$. This is generally a very weak requirement, and strictly weaker than an assumption typically implicitly made by existing PPSs when confirming the validity of their inference engines as discussed in Appendix B.

To link expectation programs back into our early expectation notation, we now note that the requirement for the return values to be a deterministic mapping of the trace means that we can write $F = f(x_{1:n})$, such that $\mathbb{E}[F] = \mathbb{E}_{\pi(x_{1:n})}[f(x_{1:n})]$. Thus the formal definition of the function we are taking the expectation of is that it is the full mapping from the raw random draws to the returned values rather than what is lexically written in any **return** statement(s). This is why, for instance, it is still valid to have multiple different **return** statements in a program; provided each **return** statement defines the same number of return values. In practice, this is not something we need to worry about when writing either models or inference engines as the law of the unconscious statistician relieves us from explicitly delineating the random variable defined by our function (the expectation of this random variable does not vary if we change the parameterization of our model). However, the distinction is important for ensuring validity and to identify the precise target function we wish to extract information about when making the inference target-aware.

## 3.2 TARGET-AWARE INFERENCE ENGINES

The key idea of our expectation programming paradigm is to use the formalisms from the previous section to set up inference engines that exploit information from $f$ to perform target-aware estimation. As explained in Section 2.2, this can lead to estimators that provide substantial performance improvements over the standard PPS approach of simply approximating $\pi(x_{1:n})$, ignoring $f(x_{1:n})$ completely.

Note that the approximate computation we are performing here is fundamentally different to that of conventional inference engines: we are estimating an expectation, rather than approximating a conditional distribution. This means the form of the outputs from our engine will change, while we will have to exploit additional information about the program. As such, we will generally need to make changes to how the program itself is processed, rather than just implementing a new inference engine in the existing PPS structure. Thankfully though, it will still usually be possible to repurpose existing inference engines as part of an overall target-aware estimation scheme, as we now show.

```
@expectation function expt_prog(y)
    x ~ Normal(0, 1)          # x ~ 𝒩(x;0,1)
    y ~ Normal(x, 1)          # y ~ 𝒩(y;x,1)
    return x^3                # f(x) = x³
end
expct_estimate, diagnostics =
  estimate_expectation(expt_prog(2),
    TABI(marginal_likelihood_estimator =
      TuringAlgorithm(AnIS(),num_samples=100)))
```

Figure 1: An example of estimating an expectation with EPT. Here **estimate_expectation** is our "do estimation" call which takes in expectation program `expt_prog` (with input $y = 2$) and an estimation method to apply (here a TABI estimator using annealed importance sampling), and returns an estimate for the expected return value of `expt_prog`.

## 3.3 EXPECTATION PROGRAMMING IN TURING

We now introduce a particular realization of the expectation programming concept which we call *Expectation Programming in Turing* (EPT). EPT builds on the PPS Turing to provide a highly effective, and surprisingly simple, mechanism to perform expectation programming. It allows users to specify $\gamma(x)$ analogously to how they would using Turing's `@model` macro, and uses Turing's **return** semantics to define $F$ and thus $f(x)$.

The key component of the EPT is splitting up the estimation of the desired expectation as per the TABI framework of Section 2.2. To do so we use source-code transformations to generate three different Turing programs, one for each of the densities $\gamma_1^+(x)$, $\gamma_1^-(x)$, and $\gamma_2(x)$ (as per Equation (3)). We then estimate the expectation by individually estimating the normalization constant of each of these densities and then combining them as per Equation (2). Generating valid Turing programs allows us to leverage any inference algorithm in Turing that provides marginal likelihood estimates to estimate the quantities $Z_1^+$, $Z_1^-$, and $Z_2$. This modularity means that we do not have to implement custom inference algorithms that would only work with EPT.

Estimating expectations with EPT is done in two stages. First, users define an expectation program with the `@expectation` macro, which is a drop-in replacement for `@model`, and an example for which is shown in Figure 1. Using code transformations, `@expectation` automatically generates the three Turing programs representing the densities $\gamma_1^+(x)$, $\gamma_1^-(x)$, and $\gamma_2(x)$. This happens behind the scenes and the user does not need to deal with the transformed programs directly.

To estimate the expectation, the user then calls **estimate_expectation**(expt_prog, method), where `method` specifies the estimation approach to be used. At present, the only supported class of methods is **TABI**, which implements the previously explained TABI estimators, but the syntax is designed to allow for easy addition of hypothetical alternative approaches.

```
@expectation function expt_prog(y)      @model function expt_prog(y)
    x ~ Normal(0, 1)                        x ~ Normal(0, 1)
    y ~ Normal(x, 1)                        y ~ Normal(x, 1)
    return x^3                              tmp = x^3
end                                         @addlogprob!(log(max(tmp, 0)))
                                            return tmp
                                        end
```

Figure 2: The results of one of the three program transformations applied to the EPT `@expectation` program from Figure 1 [left]. Presented is the transformation into a valid Turing `@model` program [right] corresponding to the density $\gamma_1^+(x) \propto \gamma(x) f^+(x)$. The transformed code fragment is highlighted. The full transformation is slightly more complex due to Turing's internals. Appendix D shows the full source code transformation for this model.

EPT then estimates the normalization constants $Z_1^+$, $Z_1^-$, and $Z_2$ by running a Turing inference algorithm on each Turing program generated by `@expectation` and combining the normalization constant estimates to form an estimate of the expectation. In the example in Figure 1, we use **TABI** with annealed importance sampling **AnIS**, which is a new Turing inference algorithm that we have added to the system for the purposes of this paper. **TuringAlgorithm** is a thin-wrapper object storing the necessary information that allows **TABI** to use a Turing inference method. **AnIS** can be substituted with any other Turing inference algorithm that returns a marginal likelihood estimate. Here **AnIS()** implies the use of some arbitrary default AnIS parameters regarding the Markov chain transition kernel, and the number and spacing of intermediate potentials used.

### 3.4 PROGRAM TRANSFORMATIONS

We now consider how to generate the Turing programs corresponding to each of the TABI densities. Note that expectation programs in EPT are also valid Turing models, i.e., replacing `@expectation` with `@model` yields a valid Turing program. Such a program corresponds to the unnormalized density $\gamma_2(x) = \gamma(x)$ without requiring any transformation of the source-code.

To create a Turing program corresponding to $\gamma_1^+(x)$, we need to multiply the unnormalized density of the unaltered Turing program $\gamma(x)$ by $\max(f(x), 0)$. This is achieved using Turing's aforementioned `@addlogprob!` primitive, such that we can think of it as adding a new factor $\max(f(x_{1:n}), 0)$ to the program density definition in (1). Our transformations are pattern matching procedures that find all the **return** expr statements in the function body and then a) create a new local variable `tmp = expr` (where `tmp` is a unique identifier generated using `gensym()`), b) insert a statement `@addlogprob!(log(max(tmp, 0)))` before the **return**, and c) change the return statement itself to **return** `tmp`. A concrete example of the transformation is presented in Figure 2. The transformation for $\gamma_1^-(x)$ is analogous but inserts a statement `@addlogprob!(log(-min(tmp, 0)))` instead.

Users can define multiple expectations by specifying multiple return values, while each individual return value

needs to almost surely be a numerical scalar. This ensures that each target expectation is well defined and individually identified. For each return expression, we apply our program transformation separately and derive a corresponding TABI estimator for each. For example, if we have **return** expr1, expr2, expr3, the program transformation for $\{\gamma_1^+(x)\}_2$ would add the statement `@addlogprob!(log(max(expr2, 0)))`. Appendix I shows a full example of this.

### 3.5 VALIDITY OF EPT

We now formalize and demonstrate the statistical correctness of the EPT approach. For simplicity, we will assume throughout that programs almost surely return a single scalar value (i.e. the probability that the return value fails to be a well-defined scalar is 0). Generalization to programs with multiple return values is straightforward (provided the number of return values is fixed) by considering each return value separately in isolation (as EPT does itself).

**Theorem 1.** *Let $\mathcal{E}$ be a valid expectation program in EPT with unnormalized density $\gamma(x_{1:n})$, defined on possible traces $x_{1:n} \in \mathcal{X}$, with return value $F = f(x_{1:n})$. Then $\gamma_1^+(x_{1:n}) := \gamma(x_{1:n}) \max(0, f(x_{1:n}))$, $\gamma_1^-(x_{1:n}) := -\gamma(x_{1:n}) \min(0, f(x_{1:n}))$, and $\gamma_2(x_{1:n}) := \gamma(x_{1:n})$ are all valid unnormalized probabilistic program densities. Further, if $\{\hat{Z}_1^+\}_m$, $\{\hat{Z}_1^-\}_m$, $\{\hat{Z}_2\}_m$ are sequences of estimators for $m \in \mathbb{N}^+$ such that*

$$\{\hat{Z}_1^\pm\}_m \xrightarrow{p} \int_{\mathcal{X}} \gamma_1^\pm(x_{1:n}) d\mu(x_{1:n}),$$

$$\{\hat{Z}_2\}_m \xrightarrow{p} \int_{\mathcal{X}} \gamma_2(x_{1:n}) d\mu(x_{1:n})$$

*where $\xrightarrow{p}$ means convergence in probability as $m \to \infty$, then $(\{\hat{Z}_1^+\}_m - \{\hat{Z}_1^-\}_m)/\{\hat{Z}_2\}_m \xrightarrow{p} \mathbb{E}[F]$.*

Theorem 1, which is proved in Appendix B, shows that if we have programs with the desired densities and we use consistent marginal likelihood estimators for each, then our resulting expectation estimates will themselves be consistent. The latter is covered by the consistency of Turing's own inference engines. The former requires that our transformed programs are valid Turing programs with the intended densities. We now show that this is indeed the case.

Given an input EPT program $\mathcal{E}$, EPT applies transformations to get the three Turing programs $\mathcal{P}_1^+$, $\mathcal{P}_1^-$, and $\mathcal{P}_2$ with $\gamma_1^+(x_{1:n})$, $\gamma_1^-(x_{1:n})$, and $\gamma_2(x_{1:n})$ as their respective densities. To ensure that the transformations for $\gamma_1^+(x_{1:n})$ and $\gamma_1^-(x_{1:n})$ are correct, we need to ensure that a) the inserted code in our transformations is itself valid, b) the transformation does not have any unintended side effects, and c) the new density terms add valid factors to the program density. The first is true as the operation of the transformed sections of code are identical to the originals except for the new `@addlogprob!` terms, which themselves produce no outputs and, by construction, use only the variables that are in scope. The second is guaranteed by ensuring that the `tmp` variables are given unique identifiers that cannot clash with each other or any other variables in the program. The third follows from the restriction that each return value must almost surely be a numerical scalar, coupled with the fact that the added density factors (namely `max(tmp, 0)` and `-min(tmp, 0)`) are non-negative by construction.

Thus, we have shown that EPT will produce a consistent estimation of program expectations, under the assumptions of Definition 4 and the consistency of the base inference algorithms implemented in Turing.

## 4 RELATED WORK

Our focus is explicitly on the case of *estimating* expectations. Though a few papers [Gordon et al., 2014, Zinkov and Shan, 2017] have provided alternative formalizations for the expectation defined by a probabilistic program, none do this from the perspective of directly targeting this expectation as the quantity to estimate. Relatedly, a few languages provide primitives to compute expectations *analytically* in the rare situation where this is possible, such as Hakaru [Zinkov and Shan, 2017] or $\lambda$PSI [Gehr et al., 2020]. Unlike in our setting, these do not require notable changes to the backend computation from the standard inference setting because the underlying problem remains the same: calculate an integral analytically. The contributions of these works are thus somewhat tangential to our own, with our key message being that *estimating* expectations *efficiently* requires a distinct computational pipeline to that of modern PPSs.

Some PPSs also provide syntactic sugars for forming expectation estimates from the samples produced by inference, but these do not adjust the inference itself to exploit target function information. For example, in Stan [Carpenter et al., 2017] users can apply target functions to posterior samples using the `generated_quantities` block. Similarly, in Pyro [Bingham et al., 2019] the return values are stored along with MCMC posterior samples, thus allowing expectations to be estimated by taking empirical averages. PyMC3 [Salvatier et al., 2016] allows users to track deterministic transformations of the latent variables. Turing itself also provides a `generated_quantities` function, similar to Stan (see Appendix C for an example).

## 5 EXPERIMENTS

We demonstrate the effectiveness of the EPT target-aware inference methods on three problems: a synthetic numerical example, an SIR epidemiology model, and a Bayesian hierarchical model. Our EPT implementation and the code for all experiments can be found at `git.io/JZOqN`.

The performance of EPT depends on the performance of the chosen marginal likelihood estimator. At the time of writing, Turing provides implementations of Sequential Monte Carlo [Del Moral et al., 2006] and Importance Sampling (IS) as inference algorithms that provide marginal likelihood estimates, but only allows using the prior as the proposal which can never be target-aware. To address this issue, we implemented a new Turing inference engine that uses Annealed Importance Sampling (AnIS) [Neal, 2001] (see Appendix A), chosen because of its ability to estimate normalization constants in high dimensions [Wallach et al., 2009, Salakhutdinov and Larochelle, 2010, Wu et al., 2017].

AnIS requires setting two hyperparameters: an annealing schedule and a transition kernel. Currently, users can choose between two transition kernels: Metropolis-Hastings (MH) implemented in `AdvancedMH.jl` [Turing Development Team, 2020] and Hamiltonian Monte Carlo (HMC) [Neal, 2011, Hoffman and Gelman, 2014, Betancourt, 2018] in `AdvancedHMC.jl` [Xu et al., 2020]. To ensure a fair comparison we use the same setup and hyperparameters for both EPT's backend and standard, non-target-aware AnIS. We also compare directly to MCMC targeting the posterior and using the same type of transition kernel as AnIS and EPT. This transition kernel is MH in Section 5.1 and HMC elsewhere. Detailed configurations are given in Appendix F.

To compare the performance of the estimators we look at the effective sample size (ESS, see below) and the relative squared error (RSE) $\hat{\delta} := (\hat{\mu} - \mu)^2/\mu^2$, where $\mu$ denotes the ground-truth value and $\hat{\mu}$ is the estimate. All our experiments correspond to target functions which are always positive, so we use $Z_1$ to refer to $Z_1^+$ as $Z_1^- = 0$. Appendix E shows how EPT can avoid computation for $Z_1^-$ when possible.

Both EPT and AnIS produce weighted samples $\{w_\ell, \hat{x}_{1:n}^\ell\}_\ell$, so we use $\mathrm{ESS}(\{w_\ell, \hat{x}_{1:n}^\ell\}_\ell) = (\sum_\ell w_\ell)^2/\sum_\ell w_\ell^2$. EPT produces two sets of samples (for $Z_1$ and $Z_2$ respectively), so we take our overall ESS as $\min(\mathrm{ESS}_{Z_1}, \mathrm{ESS}_{Z_2})$. For AnIS, we only produce one set of samples (targeting $Z_2$) but use them to estimate both $Z_1$ and $Z_2$. Here $\mathrm{ESS}_{Z_2}^{\mathrm{AnIS}}$ can be calculated in the normal way, but we have $\mathrm{ESS}_{Z_1}^{\mathrm{AnIS}}(\{w_\ell, \hat{x}_{1:n}^\ell\}_\ell) = (\sum_\ell w_\ell f(\hat{x}_{1:n}^\ell))^2/\sum_\ell (w_\ell f(\hat{x}_{1:n}^\ell))^2$. As MCMC produces unweighted samples, we cannot directly calculate analogous ESSs. Instead, we calculate an upper bound on the true ESS by assuming that the autocorrelation between samples is zero, i.e. that samples are independent. $\mathrm{ESS}_{Z_2}^{\mathrm{MCMC}}$ is then just equal to the number of samples produced, while $\mathrm{ESS}_{Z_1}^{\mathrm{MCMC}}(\{\hat{x}_{1:n}^\ell\}_\ell) = (\sum_\ell f(\hat{x}_{1:n}^\ell))^2/\sum_\ell (f(\hat{x}_{1:n}^\ell))^2$.

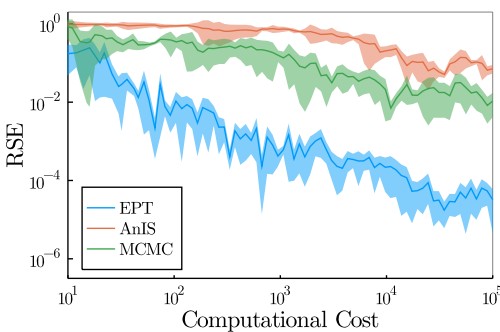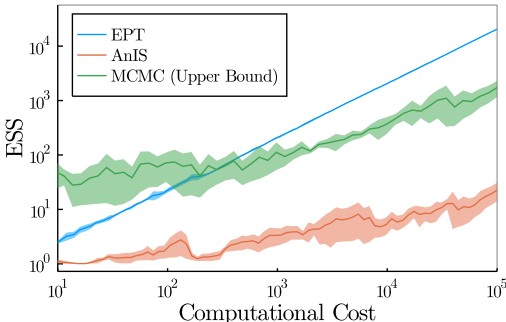

Figure 3: Relative squared error (RSE) and effective sample size (ESS) for the Gaussian posterior predictive experiment for a given computational cost. This cost is normalized across approaches by using the same number of likelihood evaluations and it has units of the combined number of samples used by EPT, which is equivalent to half the AnIS samples produced or $1/(2n)$ of the number of MCMC samples produced, where $n$ is the number of intermediary distributions used by AnIS. The solid lines show the median of the estimator while the shaded region show the 25 % and 75 % quantiles. Medians and quantiles are computed over 10 separate runs with different random seed for the posterior predictive problem. For the ESS plot we are plotting $\min(\text{ESS}_{Z_1}, \text{ESS}_{Z_2})$; note that our estimates are (quite loose) upper bounds for MCMC (see text).

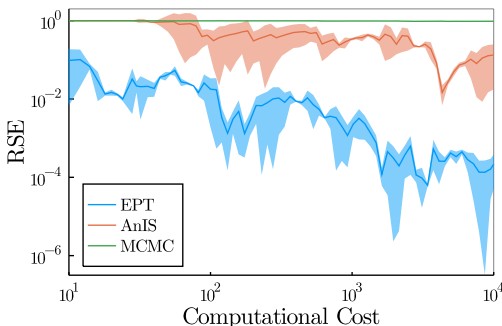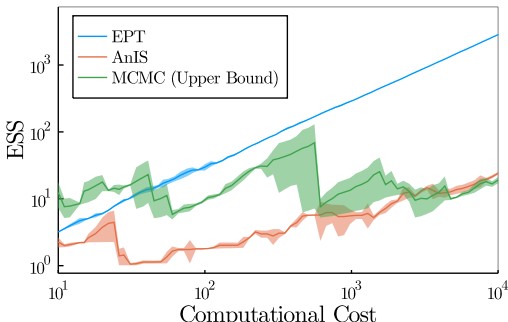

Figure 4: RSE and ESS for the SIR experiment. Conventions as in Figure 3; results computed over 5 runs.

## 5.1 GAUSSIAN POSTERIOR PREDICTIVE

The first problem considered is calculating the posterior predictive distribution of a Gaussian model with an unknown mean, where $\gamma(\mathbf{x}) = \mathcal{N}(\mathbf{x}; 0, I)\mathcal{N}(\mathbf{y}; \mathbf{x}, I)$ and $f(\mathbf{x}) = \mathcal{N}(-\mathbf{y}; \mathbf{x}, \frac{1}{2}I)$ are the unnormalised density and target function, respectively. We assume our observed data is $\mathbf{y} = (3.5/\sqrt{10})\mathbf{1}$ where $\mathbf{1}$ is a 10-dimensional vector of ones. Using EPT we can express this expectation in just 5 lines of code—the full model is given in Appendix J. This problem is amenable to an analytic solution so allows us to compute the error of the estimates. Figure 3 compares the performance of EPT, AnIS, and MCMC (here MH). We see a clear benefit to using the target-aware inference algorithm to estimate the expectation. EPT achieves a lower RSE, and the ESS highlights the advantage of using separate estimators for $Z_1$ and $Z_2$. Note that the high apparent ESS of MCMC for small sample sizes is likely due to the looseness of the bound, rather than the true actual ESS being large.

## 5.2 SIR EPIDEMIOLOGICAL MODEL

Our second problem setting is a more applied example based on the Susceptible-Infected-Recovered (SIR) model of Ker-

mack et al. [1927] from the field of epidemiology. Assume we face a disease outbreak. The government has provided us with a function yielding the expected cost of the disease which depends on the basic reproduction rate $R_0$, which indicates the expected number of people one infected person will infect in a population where everyone is susceptible. We seek to infer $R_0$ and the expected cost of the outbreak.

The SIR model divides the population into three compartments: people who are susceptible to the disease, those who are currently infected, and those who have already recovered. The dynamics of the outbreak are modelled by a set of differential equations

$$\frac{dS}{dt} = -\beta S\frac{I}{N}, \quad \frac{dI}{dt} = \beta S\frac{I}{N} - \gamma I, \quad \frac{dR}{dt} = \gamma I, \quad (4)$$

with parameters $\beta$ and $\gamma$. $S$, $I$ and $R$ correspond to the number of people susceptible, infected and recovered, respectively. The size of the total population is $N = S + I + R$. Roughly, $\beta$ models the constant rate of infectious contact between people, while $\gamma$ is the constant recovery rate of infected individuals. From these parameters we can calculate the basic reproduction rate $R_0 = \beta/\gamma$. We assume $\gamma$ to be known, and we want to infer $\beta$ and the initial number of

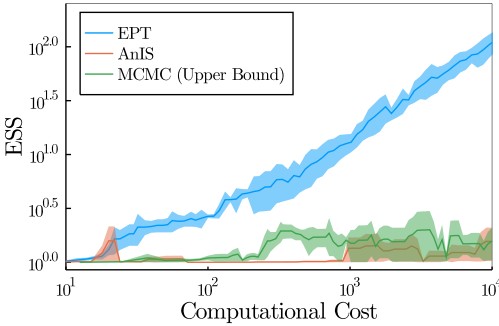

Figure 5: ESS plots for the Radon experiment. Conventions as in Figure 3; estimates based on 10 runs/seeds.

Table 1: Final estimates for the Radon experiments. Mean and standard deviation estimated over 10 runs.

| METHOD | FINAL ESTIMATE |
| --- | --- |
| EPT | 3.74e−8 ± 2.39e−9 |
| ANIS | 1.15e−9 ± 3.02e−9 |
| MCMC | 7.79e−18 ± 2.46e−17 |

infected people $I_0$. The full statistical model and the cost function (which is based on $R_0$) is given in Appendix G.

This scenario is a good use case for EPT because we are interested in estimating a specific expectation with high accuracy. Our cost function has some outcomes which might have low probability under the posterior but which incur a very high cost. These outcomes are liable to be missed by non-target-aware schemes, leading to extremely skew estimators that almost always underestimate the expectation.

Figure 4 compares the performance of the estimators. Since this problem is not amenable to an analytic solution, we estimate the ground-truth using a customized IS estimator with orders of magnitude more samples than estimates presented in the plot (see Appendix F). EPT substantially improves on the baselines, with MCMC (here HMC) failing to provide any meaningful estimate; it produces no samples where $f(x)$ is significant. EPT is able to overcome this through its use of a separate estimator for $\gamma(x)f(x)$. The fact that MCMC does far worse than AnIS, despite neither being target-aware, stems from the latter producing a greater diversity of (weighted) samples, a small number of which land in regions of high $f(x)$ by chance. To confirm that the failure of MCMC is not due to the specific implementation used we also computed results for this model in Stan, which produced similar results, see Appendix L.

### 5.3 HIERARCHICAL CONCENTRATION MODEL

Our third problem setting is a Bayesian hierarchical model for the radon concentration in households in different counties, adapted from Gelman and Hill [2006]. For the $j^{\text{th}}$ house in county $i$, we would like to predict the log radon concentration $y_{ij}$ inside the house. For each house we have a covariate $x_{ij}$ which is 0 if the house has a basement, and 1 if it does not. With this setup, the model is defined as

$$\mu_\alpha \sim \mathcal{N}(0, 10), \qquad \alpha_i \sim \mathcal{N}(\mu_\alpha, 0.12), \qquad (5)$$

$$\mu_\beta \sim \mathcal{N}(0, 10), \qquad \beta_i \sim \mathcal{N}(\mu_\beta, 0.22), \qquad (6)$$

$$\epsilon \sim \text{HalfCauchy}(0, 5), \quad y_{ij} \sim \mathcal{N}(\alpha_i + \beta_i x_{ij}, \epsilon). \quad (7)$$

We now want to find out whether the radon level in *all* households is below an acceptable level, taking this threshold to be 4pCi/L. The probability of this event is equal to the expectation under the posterior of a step function $f(x)$. However, to allow the use of HMC transition kernels we use a logistic function as a continuous relaxation of this step function. See Appendix H for more details.

This problem cannot be solved analytically and estimating the ground-truth with sufficient accuracy is computationally infeasible. We, therefore, resort to comparing EPT and AnIS based on their ESSs, noting that a low ESS almost exclusively means a poor inference estimate, while a high ESS is a strong (but not absolute) indicator of good performance. As we can see in Figure 5, EPT outperforms standard AnIS by several orders of magnitude. Additionally, Table 1 presents the final expectation estimates for each method. All methods differ in their estimates. However, EPT is the only one where the standard deviation of the estimate is small relative to its mean estimate, which, coupled with our ESS results, provides strong evidence that it is significantly outperforming the baselines. In particular, it seems clear that the MCMC (here HMC) estimate is very poor: the fact that its estimate is many orders of magnitude smaller than the others, coupled with its extremely low ESS (despite ignoring sample correlations), shows that it is failing to produce any samples in regions where $f(x)$ is non-negligible.

## 6   CONCLUSION

We have introduced the concept of expectation programming which describes the process of encoding expectations programmatically and automating their estimation in an efficient, *target-aware* manner. This concept is realized by extending the PPS Turing to EPT using a combination of program transformations and target-aware estimators. We have shown that EPT estimates expectations effectively in practice, while its modularity means that it can easily be built on by others. Moreover, we believe the introduction of the high-level expectation programming concept can pave the way for exciting future advances. While EPT focuses on the automation of TABI estimators, other implementations focusing on different approaches are conceivable—for example, systems targeting the automatic synthesis of control variates for a given input program—just as there are different PPSs focusing on distinct inference algorithms.

**Acknowledgements**

We would like to thank Sheheryar Zaidi for helpful discussions on configuring Annealed Importance Sampling. Tim Reichelt and Adam Golinski are supported by UK EPSRC CDT in Autonomous Intelligent Machines and Systems with the grants EP/S024050/1 (Tim Reichelt) and EP/L015897/1 (Adam Golinski). Luke Ong is funded by EPSRC.

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
