# OpenReview forum: "Expectation Programming: Adapting Probabilistic Programming Systems to Estimate Expectations Efficiently"
_auai.org/UAI/2022/Conference — UAI 2022 Oral_

### Official Review · Reviewer_J5vp · 2022-04-12

**Q2(1) Originality/Novelty:** 2
**Q2(2) Significance/Impact:** 2
**Q2(3) Correctness/Technical Quality:** 4
**Q2(6) Clarity Of Writing:** 3
**Q6 Overall Score:** 6
**Q8 Confidence In Your Score:** 3

**Q1 Summary And Contributions:**

This work provides a strategy to directly compute the expected values from a Turing program, by rewriting the program and providing the AnIS importance sampling inference engine. This has as advantage that the task at hand can be achieved more accurately by utilizing properties such as those presented in the TABI framework.

**Q2 Assessment Of The Paper:**

More detailed information regarding each of these aspects is given below:

**Q2(4) Quality Of Experiments (Optional):**

1: Poor: The experimental evaluation is flawed or the results fail to adequately support the main claims.

**Q2(5) Reproducibility:**

4: Excellent: Key resources (e.g., proofs, code, data) are available and key details (e.g., proof sketches, experimental setup) are comprehensively described for competent researchers to confidently and easily reproduce the main results.

**Q3 Main Strengths:**

- The focus on the task of expectations is interesting and relevant.
- Achieving this through a source transformation makes the method modular and compatible with many existing systems and approaches.

**Q4 Main Weakness:**

- The work is mainly an implementation of the TABI framework in Turing.
- The authors explain that this task, while inefficient, is available in Stan, Pyro, PyMC and plain Turing (sec 4). Why then not compare with these implementations and show the added value of the proposed method. This seems trivial to do given the programs used in the experiments and makes me wonder why this is not done. I assume that the authors consider the MCMC baseline in the experiments to be this comparison?
- There is also no comparison with just using the TABI framework. Since the paper is heavily based on that work (and implements it), this seems like a mandatory baseline.

**Q5 Detailed Comments To The Authors:**


Minor:
- 2.2.: "sometimes referred to as the marginal likelihood", wouldn't "partition function" be more common?
- Some of the sections (e.g. def 4, proof of theorem 1) seem quite straightforward.  This slows down the pace of the paper. For example, the argumentation follows directly from the definition or the assumptions part of the theorem.
- Another point where the paper build up is slow: while it is often noted that the proposed method is fundamentally different it takes a long time to show how different it is and whether approximations will be used (i.e. not fully clear whether the proposed will be an analytical method on top of the traces or an approximation when the engines are introduced in 3.2/3.3).
- "All our experiments correspond to target functions which are always positive", why this choice?

**Q7 Justification For Your Score:**

I like the overall direction of this paper but feel the comparison with existing systems is too limited. The focus is too much on the own implementation.

**Q9 Complying With Reviewing Instructions:**

1: Yes.

---

### Official Review · Reviewer_pviP · 2022-04-13

**Q2(1) Originality/Novelty:** 3
**Q2(2) Significance/Impact:** 3
**Q2(3) Correctness/Technical Quality:** 3
**Q2(6) Clarity Of Writing:** 4
**Q6 Overall Score:** 8
**Q8 Confidence In Your Score:** 3

**Q1 Summary And Contributions:**

This paper introduces the concept of “expectation programming” and develops program-transformations for Turing probabilistic programming language that turn an expectation program into several standard Turing programs, which can be used to perform inference for the expectation programming problems. The authors prove correctness of their transformation (under natural mild conditions on the program).

**Q2 Assessment Of The Paper:**

More detailed information regarding each of these aspects is given below:

**Q2(4) Quality Of Experiments (Optional):**

3: Good: The experimental evaluation is adequate, and the results convincingly support the main claims.

**Q2(5) Reproducibility:**

4: Excellent: Key resources (e.g., proofs, code, data) are available and key details (e.g., proof sketches, experimental setup) are comprehensively described for competent researchers to confidently and easily reproduce the main results.

**Q3 Main Strengths:**

Isolating an interesting sub-problem, expectation programming, for probabilistic programming is in itself an interesting contribution.

The transformation is simple, based on target-aware Bayesian inference (Rainforth et al, 2020) and I think the simplicity is really a plus here. Basically, the transformation works by pattern-matching return statements in the program and using a @addlogprob! command from Turing, which allows adding arbitrary terms to the unnormalized density function. This is nice because it allows exploiting marginal inference routines from Turing for expectation programming (however, the authors in the end also implemented a new marginal inference method into Turing: annealed importance sampling).

**Q4 Main Weakness:**

I am not an expert on PP, but I did not really see any weaknesses of this paper. Yes, it is rather straightforward, mostly exploiting target-aware Bayesian inference, but I do not think that should be a problem,

**Q5 Detailed Comments To The Authors:**

I enjoyed reading the paper and do not really have any detailed comments on how to improve it.

**Q7 Justification For Your Score:**

This is a strong paper. It identifies an overlooked problem and designs an elegant solution for it.

**Q9 Complying With Reviewing Instructions:**

1: Yes.

---

### Official Review · Reviewer_Lvva · 2022-04-18

**Q2(1) Originality/Novelty:** 2
**Q2(2) Significance/Impact:** 3
**Q2(3) Correctness/Technical Quality:** 4
**Q2(6) Clarity Of Writing:** 4
**Q6 Overall Score:** 6
**Q8 Confidence In Your Score:** 4

**Q1 Summary And Contributions:**

Computing an expectation can be more efficient if we can reformulate the task to one of estimating the normalization constants of multiple densities as shown in an earlier TABI paper.

The paper shows how the TABI idea can be easily implemented in a PPL by rewriting the program automatically. A new annealed importance sampling algorithm is mentioned but not described in the main text.

There is some theoretical justification and experiments which prove the point that TABI is a good idea.


**Q10 Ethical Concerns (Optional):**

No.


**Q2 Assessment Of The Paper:**

More detailed information regarding each of these aspects is given below:

**Q2(4) Quality Of Experiments (Optional):**

3: Good: The experimental evaluation is adequate, and the results convincingly support the main claims.

**Q2(5) Reproducibility:**

4: Excellent: Key resources (e.g., proofs, code, data) are available and key details (e.g., proof sketches, experimental setup) are comprehensively described for competent researchers to confidently and easily reproduce the main results.

**Q3 Main Strengths:**

The paper demonstrates one of the key benefits of focusing research energy on Probabilistic Programming Languages. Namely, that a current research idea in probabilistic inference such as TABI can be applied quite easily to any PPL and it can benefit all users of the PPL without them having to reimplement the idea.

Although the implementation of TABI in the target PPL of this paper is quite simple that fact should not detract from the paper. It is still a somewhat novel idea and other PPL implementors would benefit from reading this paper and learning the importance of including TABI in their own PPL,


**Q4 Main Weakness:**

The authors focused exclusively on implementing TABI and annealed importance sampling exactly as in the prior TABI paper with very similar experiments. I see this as a lost opportunity. For example, a natural question that arises to me as a PPL researcher is how does TABI + annealed IS compared to TABI + MCMC. In other words, there are MCMC algorithms that can estimate the normalization constant of unnormalized densities. Why not try any one of them and compare? Such an analysis can help inform the reader whether TABI has to be used with annealed IS or are other methods equally good with TABI.

The experiment on the SIR model shows very poor results for MCMC. This is somewhat hard to believe given that MCMC is often used in SIR models quite successfully (just google search "SIR model MCMC"), there is even an R package for this. Perhaps the authors should use a better MCMC implementation for their baseline comparison.



**Q5 Detailed Comments To The Authors:**

Beyond what is mentioned above:

Please include the ESS numbers for MCMC in all the ESS figures.

The RSE curves in Figures 3 and 4 appear to be trending downward. In this case I would be curious to know how these samplers perform with further samples, at least 10^6.

Normally most PPLs come with a library of multiple inference algorithms for estimating the normalization constant, please consider using one or more of them to help the reader get more insights on using TABI.


**Q7 Justification For Your Score:**

While the work is moderately innovative and would provide valuable insights to PPL researchers, it could benefit by comparing to more competitive baselines and other similar approaches to make this valuable.


**Q9 Complying With Reviewing Instructions:**

1: Yes.

---

### Decision · Program_Chairs · 2022-05-15

**Decision:**

Accept (Oral)

**Comment:**

Meta Review: Reviewers agree that the ideas in this paper are novel, interesting, highly relevant to UAI, and likely to have practical impact on the probabilistic programming community.  Reviewers appreciate clarifications in the rebuttal which leads to a unanimous decision to accept.